# Live-Cell Imaging of the Contractile Velocity and Transient Intracellular Ca^2+^ Fluctuations in Human Stem Cell-Derived Cardiomyocytes

**DOI:** 10.3390/cells11081280

**Published:** 2022-04-09

**Authors:** Aviseka Acharya, Harshal Nemade, Krishna Rajendra Prasad, Khadija Khan, Jürgen Hescheler, Nick Blackburn, Ruth Hemmersbach, Symeon Papadopoulos, Agapios Sachinidis

**Affiliations:** 1Working Group Sachinidis, Center for Physiology, Faculty of Medicine and University Hospital Cologne, The University of Cologne, 50931 Cologne, Germany; aacharya@uni-koeln.de (A.A.); hnemade@uni-koeln.de (H.N.); krishna.rajendraprasad@ruhr-uni-bochum.de (K.R.P.); khadijatareen1900@gmail.com (K.K.); j.hescheler@uni-koeln.de (J.H.); symeon.papadopoulos@uk-koeln.de (S.P.); 2Bioras Company, Kaarsbergsvej 2, 8400 Ebeltoft, Denmark; ndb@microwise.eu; 3German Aerospace Center, Institute of Aerospace Medicine, Gravitational Biology, Linder Hoehe, 51147 Cologne, Germany; ruth.hemmersbach@dlr.de; 4Center for Molecular Medicine Cologne (CMMC), University of Cologne, 50931 Cologne, Germany

**Keywords:** hiPSCs, contractile velocity of cardiomyocytes, CRISPR-Cas9, genetically encoded Ca^2+^-indicator, drug screening

## Abstract

Live-cell imaging techniques are essential for acquiring vital physiological and pathophysiological knowledge to understand and treat heart disease. For live-cell imaging of transient alterations of [Ca^2+^]_i_ in human cardiomyocytes, we engineered human-induced pluripotent stem cells carrying a genetically-encoded Ca^2+^-indicator (GECI). To monitor sarcomere shortening and relaxation in cardiomyocytes in real-time, we generated a α-cardiac actinin (ACTN2)-copepod (cop) green fluorescent protein (GFP^+^)-human-induced pluripotent stem cell line by using the CRISPR-Cas9 and a homology directed recombination approach. The engineered human-induced pluripotent stem cells were differentiated in transgenic GECI-enhanced GFP^+^-cardiomyocytes and ACTN2-copGFP^+^-cardiomyocytes, allowing real-time imaging of [Ca^2+^]_i_ transients and live recordings of the sarcomere shortening velocity of ACTN2-copGFP^+^-cardiomyocytes. We developed a video analysis software tool to quantify various parameters of sarcoplasmic Ca^2+^ fluctuations recorded during contraction of cardiomyocytes and to calculate the contraction velocity of cardiomyocytes in the presence and absence of different drugs affecting cardiac function. Our cellular and software tool not only proved the positive and negative inotropic and lusitropic effects of the tested cardioactive drugs but also quantified the expected effects precisely. Our platform will offer a human-relevant in vitro alternative for high-throughput drug screenings, as well as a model to explore the underlying mechanisms of cardiac diseases.

## 1. Introduction

Heart failure (HF) is a major health problem with high morbidity and mortality rate worldwide [1,2]. Inherited and non-inherited heart cardiomyopathies contribute to the development of HF too [3,4]. Persistent cardiac architectural changes of the sarcomere cytoskeletal cardiac proteins may cause dilated cardiomyopathy and, therefore, HF [3,4]. It is well established that dysfunction of cardiac contractility and relaxation due to sarcomere defects in the contractile apparatus, in combination with changes in intracellular Ca^2+^ homeostasis, are hallmark characteristics of HF and, therefore, ideal therapeutic targets [5,6]. Cellular cytoskeletal proteins are essential for contributing to the cell intact function [7,8]. In particular, the sarcomere contraction/relaxation kinetics are of fundamental importance to the development and progression of HF [6]. It is also well established that anticancer drug-induced cardiotoxicity leads to HF.

Human-induced pluripotent stem cells (hiPSCs) have proven to be an efficient model to study novel pathological mechanisms of genetic diseases and provide an unlimited source for generating somatic cells for cellular therapy of degenerative diseases [9]. In this context, one of the main advantages of using hiPSCs and their derivatives is their human origin, which significantly minimizes physiological differences that normally exist between humans and other animal species [10,11]. Moreover, hiPSCs and cardiomyocytes (CMs) derived from hiPSCs (hiPSC-CMs) are proven as an efficient drug discovery model for safety pharmacology [10,11]. Several hiPSC-CM-based in vitro assay platforms for pharmacological screening have been described [11,12,13]. In this context, several genomic biomarkers, characteristic for cancer drug-induced cardiotoxicity in hiPSC-CMs, were identified [14,15,16,17,18].

In recent times, more advanced technologies, like video-based imaging and fluorescence microscopy, in combination with hiPSC-CMs, have been developed for drug cardiotoxicity screening and investigating mechanisms underlying cardiotoxicity [19,20,21,22]. However, technologies allowing live-cell imaging of the contractile functional parameters, such as sarcomere length, contraction/relaxation velocity, and the Ca^2+^ homeostasis, remain a challenge [10,11].

The classical non-live imaging procedures to monitor changes on the molecular level (e.g., changes in the cytoskeleton) are inefficient, static methods, executed on fixed cells. For more efficient (cost-effective and less time consuming) monitoring of the dynamic changes of molecular parameters, such as the cytoskeleton or of intracellular free Ca^2+^ ([Ca^2+^]_i_) under different pathological or pharmacological conditions, live-cell imaging of these molecular parameters is needed. Live-cell imaging also delivers more physiologically relevant information in comparison to fixed cell microscopy.

The clustered regularly interspaced short palindromic repeats (CRISPR) and associated (Cas)-9 technology efficiently facilitates genetic editing in human pluripotent stem cells, including hiPSCs and human embryonic stem cells (hESCs), thereby dramatically increasing the potential applications of hiPSCs as a disease model for monogenic, polygenic, and rare diseases, very often associated with gene mutations [9]. Thus, the impact of the CRISPR/Cas9 technology opens several avenues of research into the underlying mechanisms and therapy of cardiovascular diseases. Moreover, this technology in combination with hiPSC-CMs allows comprehensive cardiac research to identify drug-induced cardiotoxicity (for review see [23]). Here, we generated two transgene hiPSC cell lines allowing the production of hiPSC-CMs, enabling live-cell imaging of the sarcomeres and transient changes of [Ca^2+^]_i_. Our novel video analysis software also allowed us to quantify functional parameters related to contraction and relaxation. We validated the applicability of our platform using Isoprenaline as a classic Adrenoreceptor agonist, the L-type agonist Bay-K8864, the L-Type blocker Nifedipine, and the muscarinic agonist Carbachol. We firmly believe that these sophisticated tools open new horizons to study the contractility of CMs under pathological conditions. This technology can be applied to screen pharmacological drugs and to study the mechanisms involved in the development of cardiac diseases.

## 2. Materials and Methods

### 2.1. Differentiation of hiPSCs to Cardiomyocytes

Experiments were performed with the IMR90 hiPSCs (authorized by the Robert-Koch Institute; Berlin, Germany, license number: AZ 3.04.0210083). Cells were cultured on matrigel-coated petri dishes. Cells were cultured in StemMACS™ iPS-Brew XF media (Milteny Biotech, Bergisch Gladbach, Germany) supplemented with 50 U/mL penicillin and 50 U/mL streptomycin (Thermo Fisher, Waltham, MA, USA), at 37 °C and 5% CO_2_. After reaching a confluence of 90%, cells were dissociated by trypsinization using the trypLE solution (Thermo Fisher, Waltham, MA, USA) and propagation of the cells was performed in StemMACS™ iPS-Brew XF media in the presence of 10 µM ROCK inhibitor. Differentiation of hiPSCs to CMs was performed using a combination of two differentiation protocols [24,25] as described previously [26]. Briefly, culturing of the hiPSCs up to 70% confluence was performed in StemMACS™ iPS-Brew XF media (day 0). Then, cells were cultured in Roswell Park Memorial Institute (RPMI; Thermo Fisher, Waltham, MA, USA) RPMI1640/B-27 without insulin in the presence of 10 µM of 6-[[2-[[4-(2,4-Dichlorophenyl)-5-(5-methyl-1H-imidazol-2-yl)-2-pyrimidiyl]amino]ethyl]amino]-3-pyridinecarbonitriletrihydrochloride CHIR99021 (Tocris, Bristol, UK) for 24 h (day 1). The culture medium was replaced with normal RPMI1640/B-27 without insulin and cells were cultured for further 24 h (day 2). Again, the culture medium was replaced with RPMI1640/B-27 without insulin, containing 10 µM of *N*-(6-Methyl-2-benzothiazolyl)-2-[(3,4,6,7-tetrahydro-4-oxo-3-phenylthieno [3,2-d]pyrimidin-2-yl)thio]-acetamide IWP2 (Tocris, UK) and cells were cultured for further 48 h (day 4). The medium was replaced with basal RPMI1640/B-27 without insulin and was refreshed every 48 h. Spontaneous beating in cardiomyocytes was observed from day 9 onwards. Enrichment of the cardiomyocytes was performed by culturing of the cells with glycose-free Roswell Park Memorial Institute (RPMI; Thermo Fisher, Waltham, MA, USA) containing 4 µM sodium DL-lactate for 6 d. The efficiency of the differentiation of hiPSCs to obtain a purity of CMs over 95% by flow cytometry and RT-qPCR and hiPSC-CMs was characterized using immunocytochemistry, electrophysiology, electron microscopy, and calcium transient measurements [26]. In all our experiments, the hiPSC-CMs were cultured for longer durations of up to 30 d. The hiPSC-CMs were trypsinized and seeded into different plastic materials, which were pre-coated with fibronectin. Seeding of the cells was performed in Dulbecco’s Modified Eagle Medium (DMEM) containing 5% foetal calf serum. The hiPSC-CMs were cultured in a modified cardiomyocyte maturation medium [27] in a 5% CO_2_ incubator at 37 °C. The modified maturation medium was composed of glycose-free RPMI medium supplemented with 4 mM of (2S)-2-hydroxypropanoate L-lactate (Sigma Aldrich, St. Louis, MO, USA), 5 mM of 2-(1-Methylguanidino)acetic acid hydrate creatine monohydrate (Sigma Aldrich, St. Louis, MO, USA), 2 mM of 2-aminoethanesulfonic acid taurine (Sigma Aldrich, T0625), 2 mM of (3R)-3-Hydroxy-4-(trimethylammonio)butanoate L-carnitine (Sigma Aldrich, St. Louis, MO, USA), 0.5 mM of (5R)-[(1S)-1,2-Dihydroxyethyl]-3,4-dihydroxyfuran-2(5H)-one ascorbic acid (Sigma Aldrich, St. Louis, MO, USA), 1x Linoleic Acid-Oleic Acid-Albumin 100x (Sigma, St. Louis, MO, USA), Non-Essential Amino Acid (NEAA) (Thermo Fisher, Waltham, MA, USA), 1x B27 media, and 1% Knockout serum replacement (KOSR) (Thermo Fisher, Waltham, MA, USA). The seeded hiPSC-CMs were cultured for 48–72 h to obtain the uniform beating pattern in the flasks prior to the experiments.

### 2.2. Generation of the GECI-eGFP^+^-hiPSCs

For live-cell imaging of transient alterations of the [Ca^2+^]_i_ in CMs, we generated genetically-encoded Ca^2+^-indicator (GECI)-enhanced green fluorescent protein (eGFP^+^)-hiPSCs (IMR90), as indicated in Figure 1. The transgenic GECI-eGFP^+^-hiPSCs were then differentiated to GECI-eGFP^+^-CMs for further experiments.

In brief, first the *CmR* gene from pB-TAC-ERP2 plasmid was replaced with the *GCaMP6s* gene using LR clonase. Next, the DNA construct was confirmed by DNA sequencing and then transfected into the hiPSCs along with the pCAGPBase plasmid. These transfected cells were cultured in the presence of 2 µg/mL puromycin and positive clones (generated clones labeled as generation of the genetically encoded Ca^2+^ indicator (GECI-eGFP^+^-hiPSCs) were selected and expanded. Then, we differentiated the hiPSCs to transgenic GECI-eGFP^+^-CMs, allowing live-cell imaging of alterations of [Ca^2+^]_i_ during contraction of CMs. Expression of GECI in CMs was induced by adding 500 nM of (4S,4aR,5S,5aR,6R,12aR)-4-(dimethylamino)-1,5,10,11,12a-pentahydroxy-6-methyl-3,12-dioxo-4a,5,5a,6-tetrahydro-4H-tetracene-2-carboxamide;ethanol;hydrate;hydrochloride Doxycycline (). The binding of Ca^2+^ to GECI results in a conformational change of CaM, thereby binding to target proteins, such as M13, and enhancing the fluorescence signal of eGFP (by de-protonation of the eGFP chromophore). In the absence of Ca^2+^, the eGFP chromophore is protonated and the fluorescence intensity is very poor. Plasmids and the gateway adapter sequences required for generation of GECI-eGFP+-hiPSCs are shown in Appendix A.

### 2.3. Generation of ACTN2-copGFP^+^-hiPSC Line

To monitor CM sarcomeres contraction and relaxation in real-time, we generated an ACTN2-copGFP^+^-hiPSC line using the CRISPR-Cas9 and the homology-directed recombination approach, as indicated in the Figure 2. 

In brief, the ACTN2-gRNA was cloned into the pX330 plasmid and confirmed by DNA sequencing. Next, ACTN2 homology arms were cloned in the universal donor vector and confirmed by DNA sequencing. Then, the donor and gRNA plasmids were transfected in the hiPSCs and positive clones were selected using 2 µg/mL puromycin. The DNA sequences of the ACTN2-gRNA with the PAM sequence (underlined), the donor plasmid with the left and right 600 bp 5′ and 3′ homology arms (LHA and RHA, respectively), and the Gly-linker are shown in Appendix A. Lastly, we differentiated the hiPSCs to transgenic ACTN2-copGFP^+^-CMs for live-cell imaging.

### 2.4. Video Analyzer 1.9

We developed a software tool to image live changes of [Ca^2+^]_i_ during contraction of CMs, as well as the contractile sarcomere activity of the CMs. The software is based on the LabVIEW (ni.com: https://www.ni.com/de-de.html, accessed on 3 March 2022) application, where we have programmed the Vision Module to monitor and record fluorescence signals of microscopic live-cell imaging records. The software can be used to calculate the contraction/relaxation velocity of the CM sarcomeres, as well as changes in [Ca^2+^]_i_, based on video-image recordings. All videos were recorded using a frame rate of 50 fps. The velocity field was calculated for each frame, using the Horn and Schunck algorithm [28]. The velocity points of the top 1/3000 fraction of the whole velocity field were used for further analysis. The signal processing for the sarcomere contraction/relaxation velocity was performed as follows: (1) Only peaks greater than 5% of the maximum peak level above the baseline in the respective velocity distributions were included; (2) The baseline was defined at the 10% percentile; (3) Peaks were matched into pairs, when the end of the first peak was close to the beginning of the second peak. A pair was considered to represent a contraction followed by a dilation; (4) The start and endpoints of a peak were selected to be the points 20% above the baseline level on each side of the peak; and (5) The delay between the end of a contraction and the start of dilation was taken as the interval between the end of the first peak and the start of the second peak.

The calculation of the velocity of the intracellular changes of Ca^2+^ was performed after subtraction of the background on color video images. The average intensity of the green image frames was normalized to the maximum of 255 and calculated excluding zero-pixel values. The difference in intensity between successive frames was leveraged to define transient Ca^2+^ sparks in fluorescence. The difference was thresholded to a fixed minimum value of 10, and the average intensity was calculated by normalizing values to a maximum of 255 (Sp_Intensity). The relative area that the sparks occupied in the whole image was calculated (Sp_Area). The signal processing was performed as follows: (1) The baseline was set at the 10% percentile, the peak widths of the green pulses were set at 10% height above the baseline, and the peak heights were set relative to the baseline; (2) The spark event count before each green peak was calculated from the number of peaks in the spark intensity data; and (3) The steepness (ΔF/Δt)_max_ was calculated as the largest increase in intensity on the rising edge of each peak within a video frame (ΔT).

All image analyses technology described in this study were developed in LabVIEW (www.ni.com, accessed on 1 December 2021). Modification of the code requires a full development version of LabVIEW 2012 or newer as well as the NI Vision Development Module. Although the code is distributed in this study for users to further modify, under GNU license, users who simply wish to apply the code can do so without programming knowledge using the supplied application, which can be freely installed https://github.com/nblackburn123/Video-Analyser/releases/tag/v1.9 (accessed on 4 March 2022) on any Windows operating system (if you use the software please cite this article).

### 2.5. Statistical Analysis

For all experiments, the statistical errors were represented as mean ± standard error of the mean (SEM). To calculate the *p*-value of significance, Two-tailed Student’s *t*-tests or ANOVA were used and *p* values ≤ 0.05 were considered statistically significant.

## 3. Results

### 3.1. Live-Cell Imaging of Intracellular Ca^2+^ Alterations during Contraction of GECI-eGFP^+^-CMs

Imaging of eGFP fluorescence waves related to Ca^2+^ in beating CMs was initiated by the addition of Doxycycline (500 nM) for 6 h (Figure 3A). Representative video recordings (recorded with imaging frame rates of 50 fps) of Ca^2+^ dynamics in control GECI-eGFP^+^-CMs (referred to as c-CMs) are shown in Appendix A. As indicated, the maximal [Ca^2+^]_i_ occurred after 0.1 s and the (time to 90% of peak) T_90_ time was approximately 0.4 s. The video records of the [Ca^2+^]_i_ fluctuation curves were automatically analyzed using the software Video Analyzer 1.9 (VA1.9; for further details see Materials and Methods). Appendix A shows representative analyses of the regular transient fluctuations in [Ca^2+^]_i_ waves of fluorescence in c-CMs during contraction in the absence (end concentration of Dimethylsulfoxid: 0.05%) and presence of Bay-K8864, Isoprenaline, Nifedipine, and Carbachol (end concentration of Dimethylsulfoxid: 0.05%). Temporal parameters of the [Ca^2+^]_i_ fluorescence signals are shown in Figure 3B.

T_0_ represents the time point immediately before a rise in [Ca^2+^]_i_. Time-to-peak (TTP) represents the time required for the transient to reach a maximum (Figure 3A). T_90_ represents the time required for the fluorescence signal to return from its maximum to 10% of the amplitude. We calculated the maximum slope of the rise in [Ca^2+^]_i_, using (∆F/∆T)_max_; this value was used as indicator for inotropic effects (Figure 3A). Normalization of the six different, independent experiments was performed by expressing (∆F/∆T)_max_ and T_90_ values, calculated for experiments where different agents had been used, as percentages of the values calculated for c-CMs, whereby the latter were set to 100%. A representative video recording after stimulation of the c-CMs with 1 µM of adrenoreceptor agonist [1-hydroxy-2-(propan-2-ylamino)ethyl]benzene-1,2-diol;hydrochloride Isoprenaline (Sigma Aldrich, St. Louis, MO, USA) and subsequently with 1 µM of L-Type channel blocker dimethyl 2,6-dimethyl-4-(2-nitrophenyl)-1,4-dihydropyridine-3,5-dicarboxylate;hydrochloride Nifedipine (Sigma Aldrich, St. Louis, MO, USA) is shown in Appendix A. As seen in the video, the velocity of the Ca^2+^-waves was significantly increased by Isoprenaline treatment; while, as expected, addition of Nifedipine not only reduced the velocity, but also the resting sarcoplasmic Ca^2+^ level (CM beating also stopped). A representative video recording of the control (c-CMs) followed by addition with Isoprenaline (1 µM) and subsequently by addition of 1 µM of acetylcholine receptor agonist 2-carbamoyloxyethyl-trimethyl-azanium Carbachol (Sigma Aldrich, St. Louis, MO, USA) is shown in Appendix A. Again, Isoprenaline significantly increased the velocity of Ca^2+^-fluorescence waves, while Carbachol significantly reduced the velocity of the Ca^2+^ fluctuations as well as the Ca^2+^ fluorescence correlated with high [Ca^2+^]_i_. A representative video recording of the c-CMs treated with the L-type calcium channel agonist 1 µM of (±)-1,4-Dihydro-2,6-dimethyl-5-nitro4-(2-[trifluoromethyl]-phenyl)pyridine-3-carboxylic acid methyl ester Bay-K8864 (Sigma Aldrich, St. Louis, MO, USA) is shown in Appendix A. As expected, Bay-K8864 drastically increased the velocity of the fluctuations of Ca^2+^-fluorescence. As indicated in Figure 3B, Isoprenaline and Bay-K8864 caused an 51% and 30% increases in (∆F/∆T)_max_ over the values for c-CMs (= 100%), respectively. The addition of Carbachol or Nifedipine to the Isoprenaline-stimulated c-CMs resulted in a significant reduction of (∆F/∆T)_max_ compared with the c-CM value, reducing these values from 100% to 59% and 19%, respectively. Isoprenaline alone resulted in the reduction of the T_90_ value, compared with that of the c-CM value, from 100% to approximately 78%, whereas Bay-K8864 induced a 20% increase in the T_90_ value over the c-CM value.

### 3.2. Live-Time Imaging of the Contractile Velocity of ACTN2-copGFP^+^-CMs in the Presence or Absence of Different Agonists and Antagonists

Generation of the *α-actinin (ACTN2)*-copepod (cop) GFP^+^-hiPSCs was performed as described in Figure 2, and transgenic cells were differentiated to CMs as described in the Material and Methods.

Figure 4 shows the Z-discs of ACTN2-copGFP^+^-CMs (ACTN2 is enriched in Z-discs of the sarcomeres of CMs; see also Appendix A).

The video recordings of the fluctuations of the contraction and relaxation velocity were analyzed using the software: VA1.9 (for representative fluorescence contraction-relaxation cycles, see Appendix A). All videos were recorded at 50 fps rate. Figure 4A shows one representative contraction-relaxation cycle divided into contraction and relaxation phases. As indicated, the program allowed us to determine all experimental parameters between time points 1 to 5 in Figure 4A. The key parameters, such as TTP, (∆F/∆T)_max_, and the durations of contraction and relaxation, as well as the beating frequency, were calculated. Values are expressed as percentages of the c-CM values, which were set to 100%. Appendix A show that the contractile velocity of the c-CMs after stimulation with Isoprenaline and Bay-K8864, was significantly increased.

As indicated in Figure 4B, after stimulation of the c-CMs with Bay-K8864 both contraction and relaxation phases were slightly decreased to 88%, compared to c-CM values, although this effect was not statistically significant. Stimulation of the c-CMs with Isoprenaline caused a 37% and 25% inhibition of the contraction and relaxation phases, respectively. Subsequent addition of Carbachol almost recovered the inhibitory effect of Isoprenaline for the contraction phase but reduced the relaxation time of Isoprenaline to 40% of the c-CM value induced by Isoprenaline. As shown in Figure 4B, Isoprenaline and Bay-K8864 treatments resulted in an increase in the CM beating rate by 2.2- and 1.4-fold, respectively, compared to that of c-CMs. In contrast, consecutive treatments with Isoprenaline and Carbachol resulted in only a 1.5-fold increase in the CM beating rate, indicating that Carbachol, to some extent, inhibited the Isoprenaline-induced increase in the beating rate.

## 4. Discussion

There is no doubt about the superiority of non-invasive live-cell imaging methods over classical methods for investigating cellular and intracellular biological processes. Given their advantages, live-cell imaging techniques were recently applied in several disciplines, including biomedicine, cell biology, pharmacology, and developmental biology, to obtain more reliable results than from fixed cells and tissues (for review see [29,30]). Live-cell imaging techniques were used for real-time investigation of intracellular structures and cellular processes. These real-time techniques not only enabled visualization of intracellular organelles, e.g., mitochondria, in real-time but also facilitated the study of dynamic processes in cells under conditions of physiological, pathophysiological and drug toxicity. In general, these techniques produced more reliable and physiological findings in comparison to fixed cell microscopy (for review see [30]).

The application of single fluorescent protein-based genetically encoded biosensors is based on allosteric modulation of the fluorescence of a single fluorescent protein [31]. An example of such a genetically encoded biosensor is the Ca^2+^ biosensor/GCaMP biosensor, which has been successfully applied in neurosciences for imaging of neuronal activities [31]. Moreover, also other groups generated biosensor/GCaMP hiPSC-CMs [32,33,34]. However, our study differs significantly in the way that the GCaMP donor vector is created and expressed in the hiPSCs. With the use of our construct, the user can precisely control the expression of the GCaMP using Tet-on system. We also opted for directional insertion of the GCaMP in AAVS1 site. Therefore, an advantage of our genetically encoded Ca^2+^ biosensor/GCaMP hiPSC-CMs is the real-time monitoring of transient fluctuations of [Ca^2+^] in a Tet-on inducible manner.

Multiple advantages of using genetically encoded biosensors for Ca^2+^-imaging (based on protein fluorescence, such as eGFP) over synthetic dyes, such as fluo-2, are outlined below. Clearly, in comparison to new technologies, the loading procedure of Ca^2+^ indicators, such as fluo-2, is invasive and time-consuming [35]. In addition, control of the cellular permeabilization and concentration of the cytosolic Ca^2+^ fluorescent dyes is limited [36]. In contrast, changes in intracellular Ca^2+^ in GECI CMs are evident in all CMs and do not depend on the variances of the uptake of Ca^2+^ dyes in CMs [37,38]. The genetically encoded biosensor cell lines, such as the GECI hiPSCs, can be easily distributed to different laboratories. Therefore, findings with the GECI hiPSC-derived CMs will be more robust and reliable, compared to invasive and time-onsuming experiments with fluo-2 and other similar Ca^2+^ dyes [31]. Another major drawback of using such dyes, if not the most important, is that experiments with fluo-2 (and fluo-4, which we have used extensively in our lab) are final, i.e., the cells are harmed severely by subcellular accumulation, or even precipitation, of the fluorophore in a time-dependent manner (personal observations), so that cells cannot be re-used, e.g., in their latter stage of development. It is also known that fluorescent Ca^2+^ dyes exert off-target effects by inhibiting ATPases. In contrast, long-term recording of the fluctuations of the Ca^2+^ in GECI CMs is possible without inducing deleterious off-target effects [39,40]. In this context, monitoring of Ca^2+^ in GECI CMs can be controlled by switching the detection on or off.

More recently, the CRISPR/Cas9 technology was used as an efficient technology for tagging transcriptionally silent endogenous genes in hiPSCs, enabling live visualization of cytoskeletal proteins. In this context, an eGFP fusion tag and a constitutively expressed mCherry fluorescence selection cassette were delivered via homology directed repair to the endogenous cardiac cytoskeleton specific genes, such as *ACTN2* into hiPSCs, allowing live imaging of sarcomeres after differentiation of the hiPSCs to CMs [41].

To date, several gene editing lines have been reported applying established techniques [41,42]. However, we believe that the key advantage of our line is that we have edited the native *ACTN2* gene by removing its stop codon and replacing it with copGFP. This is very different from many other reported lines generated using traditional methods, because they use the promoters from either *ACTN2* or *MYH* genes to express copGFP. Thus, the traditional approach incorporates excess DNA material at random loci in the cellular genome DNA that may lead to inconsistent results.

There are several other recently developed software’s to analyse the contractility of hiPSCs-derived CMs [7,43,44,45,46].

For example, software’s like SarcTrack [46] and CalTrack [44] use MatLab-based algorithms which needs expertise in setting up the software’s as well as high-end computing systems to process the MatLab data sets. Whereas softwares like Musclemotion [45] and SarcOptim [43] are relatively easy to use but needs imageJ-based plugins which the user has to download and install. In contrast, our user-friendly interface allows any user to upload and analyse the data with normal computer set-ups without needing any software code optimization or need to install any specific plugins. In addition, when we used SarcTrack in our analysis, we noticed several limitations. For example, many times the software fails to recognize the true sarcomere bands and, hence, cannot follow the sarcomeric contractions properly causing program to re-plots the sarcomeric bands with incorrect readings/data. Another advantage of our software is that it is designed to analyse both sarcomeric contractions and calcium flux in CMs as a 2-in-1 system whereas with the other systems are designed to perform single function analysis for example, SarcTrack can only do sarcomere length measurements and the CalTrack can only measure calcium flux in CMs.

The contractility of CMs is regulated by [Ca^2+^]_i_. The binding of Ca^2+^ to troponin C, a protein of the troponin complex (troponin C, troponin I, and troponin T) initiates contraction of the sarcomeres and can be visualized as a shortening of the distance between Z-discs (systolic phase). Dissociation of Ca^2+^ from troponin C initiates the relaxation of CMs, which can be monitored by the increase in the Z–Z distance. (diastolic phase). To investigate the dynamic changes of the contractility of CMs, pending on the [Ca^2+^]_i_, we also generated a transgenic hiPSC cell line, allowing live-imaging of the transient changes in [Ca^2+^]_i_. The applicability of the transgenic hiPSC-CMs as a cellular platform for live imaging of beneficial or adverse functional effects of different drugs was tested using the sympathomimetic drug, Isoprenaline, and the parasympathomimetic drug, Carbachol, as well as an agonist Bay-K8864 and antagonist Nifedipine of the L-type channel. In parallel, we developed a 2-in-1, user friendly software VA1.9 to quantify the contraction and relaxation kinetics, as well as the kinetics of transient fluctuations of [Ca^2+^]_i_ during contraction and relaxation phases of the CMs.

Bay-K8864 is a potent inotropic compound that increases the action potential duration (APD) in adult animal ventricular heart muscles and Purkinje fibers [47]. It is established that the Bay-K8864-induced Ca^2+^-influx via the L-type channel induces a Ca^2+^-release from the sarcoplasmic reticulum [48] (Ca^2+^-induced Ca^2+^-release mechanism). CMs from mouse ESCs showed prolonged APD [49] and Bay-K8864 induced a 27% increase of the APD_90_ in ventricular human ESCs-derived CMs [22], as well as in hiPSC-derived CMs [20]. Similarly, we demonstrated that Bay-K8864 slightly but significantly increased the T_90_ period that correlates positively with the APD_90_. Moreover, we demonstrated that Bay-K8864 increased (∆F/∆T)_max_, indicative of an inotropic effect. In a comparative electrophysiological study in which several positive inotropic agents were tested on hiPSC-CMs, it has been shown that Isoprenaline slightly decreased APD_90_, whereas Bay-K8864-induced a prolongation of the APD_90_ [19]. Isoprenaline (alias isoproterenol) is a positive inotropic, lusitropic and chronotropic drug, acting by binding to the β1-Adrenoreceptors of cardiac cells (ventricular, atrial, and pacemaker) and elevating the intracellular cAMP level, thereby activating protein kinase C [19,50]. In another study, Isoprenaline shortened the APD_90_ of hiPSC-CMs [21]. Our findings show positive inotropic effects of Isoprenaline that significantly elevated the (∆F/∆T)_max_, and significantly shortened the T_90_ period, which correlates well with ADP_90_ values. Carbachol not only abolished the effect of Isoprenaline on the (∆F/∆T)_max_, and on the T_90_ value, but also significantly reduced the (∆F/∆T)_max_ value, even below that of the c-CM value.

Our findings with the ACTN2-copGFP^+^-CMs demonstrated that both Bay-K8864 and Isoprenaline elevated (∆F/∆T)_max_ values, again proving the inotropic effects of both drugs. As expected, Carbachol almost eliminated the effect of Isoprenaline on the basal values of c-CMs. Moreover, Isoprenaline induced a significant shortening of the contraction and relaxation phases, as observed by other authors in rat, mouse, and human engineered heart tissues [51]. The lusitropic effect of Isoprenaline was not observed in hiPSC-CMs obtained by Cellular Dynamics (iCell International, Madison, WI). According to these authors, the lack of the lusitropic effect of Isoprenaline might be explained by a low expression of phospholamban in these hiPSC-CMs [19]. Our data confirm that there are inotropic and lusitropic effects of Isoprenaline on heart tissues and CMs. No significant effect was observed for Bay-K8864 on the contraction and relaxation phases of the c-CMs. As shown in Figure 4, our findings are compatible with the beating frequencies of c-CMs in the presence of Isoprenaline and Bay-K8864, both of which significantly increased the beating rate, although the effect of Isoprenaline was partly abolished by Carbachol (Nifedipine completely inhibited the beating activity of c-CMs). In conclusion, our live-imaging platform is applicable to screening and testing of potential drugs and toxicants on human cardiomyocyte function. Moreover, our platform will contribute to better understanding of the underlying mechanisms of the development and therapy of heart diseases.

## Figures and Tables

**Figure 1 cells-11-01280-f001:**
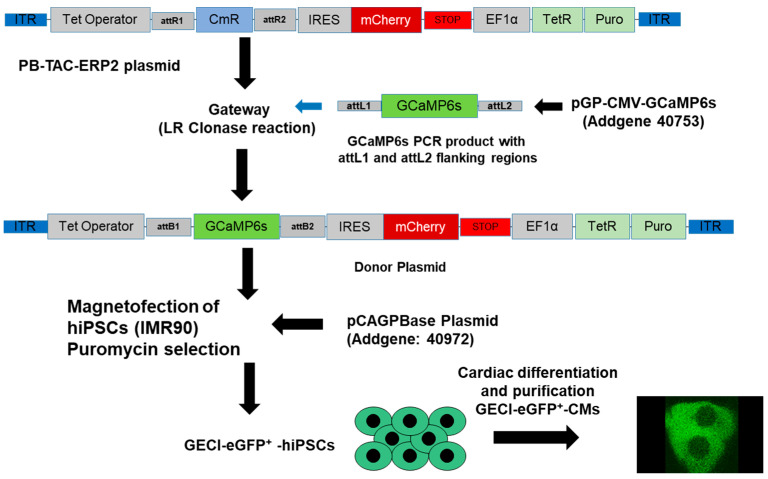
Generation of Tet-inducible GECI-eGFP^+^-hiPSCs -enhanced green fluorescent protein (eGFP^+^)-human-induced pluripotent stem cell (hiPSCs) for live calcium flux measurements in hiPSC-cardiomyocytes (CMs). GCaMP6 is a GECI generated from a fusion of the eGFP, calmodulin (CaM), and a short peptide from myosin light chain kinase (M13). To generate the Tet-inducible GECI plasmid, we amplified the GCaMP6s sequence from pGP-CMV-GCaMP6s (40753; Addgene, Watertown, MA, USA) by adding a gateway adapter sequence at both 5′ and 3′ end. The amplified GCaMP6s was cloned into the Piggybac (PB), PB-TAC-ERP2 vector (80478; Addgene, Watertown, MA), using the Gateway™ LR Clonase™ II enzyme mix (Thermo Fisher, Waltham, MA, USA). The hiPSCs were transfected with the PB-TAC-ERP2-GCaMP6s plasmid (Donor plasmid; Addgene, Watertown, MA, USA) and the piggybac transposase vector (pCAGPBase; Addgene, Watertown, MA, USA) applying the magnetofection method (Magnetofectamine O2, OZ Biosciences, Marseille, France) to generate the GECI hiPSC line. The transposase enzyme facilitates the integration of the DNA elements in ITR sites present in the genome at random location The selection was performed with puromycin at a concentration of 2 µg/mL. After generation of the GECI-eGFP^+^-hiPSCs, cells were differentiated into GECI-eGFP^+^-CMs. Induction of GECI in CMs was induced by adding Doxycycline (500 nM) for 6 h (attL-recombination site left, attR-recombination site right, attB-attachment site bacteria).

**Figure 2 cells-11-01280-f002:**
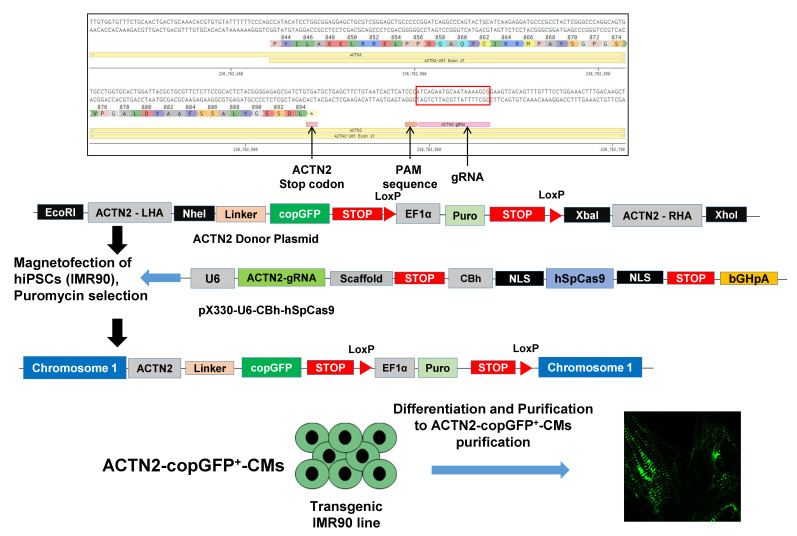
Generation of α-actinin (ACTN2)-copepod green fluorescent protein (copGFP^+^)-human-induced pluripotent stem cell line (IMR90) by the CRISPR-Cas9 and the homology-directed recombination (HDR) approach. The overall CRISPR-Cas9-based strategy to generate the transgenic ACTN2-copGFP-IMR90 line was by copGFP knock-in at chromosome. First, gRNA targeting of the ACTN2 3′ end was designed to delete the stop codon in exon 21 of the native ACTN2, located into chromosome 1. The donor plasmid with ~600 bp 5′ and 3′ homology arms was designed and ordered from ALSTEM, LLC San Francisco, US. The homology arms were designed in such a way that upon HDR the stop codon from the native *ACTN2* gene could be deleted, whereas the reading frame remained as it was. These homology arms were then cloned into the pUC57 plasmid backbone along with the copGFP and LoxP flanked puromycin resistance gene with the EF1α promoter. To avoid any interference from copGFP with the ACTN2 functions and vice-versa, a glycine-rich linker (GGGGSGGGGSGGGGS) sequence was added. The linker provided a flexible connection between the two proteins, while avoiding interference in their functional properties (cloning sequences are shown in the Appendix A). The donor plasmid along with ACTN2-gRNA plasmid was transfected into IMR90 cells using the magnetofectamine method (OZ Biosciences, Marseille, France) and positive clones were selected using puromycin (2 µg/mL). Positive clones were expanded and knock-in of copGFP was confirmed by differentiating the cells into cardiomyocytes by following our standard differentiation protocol.

**Figure 3 cells-11-01280-f003:**
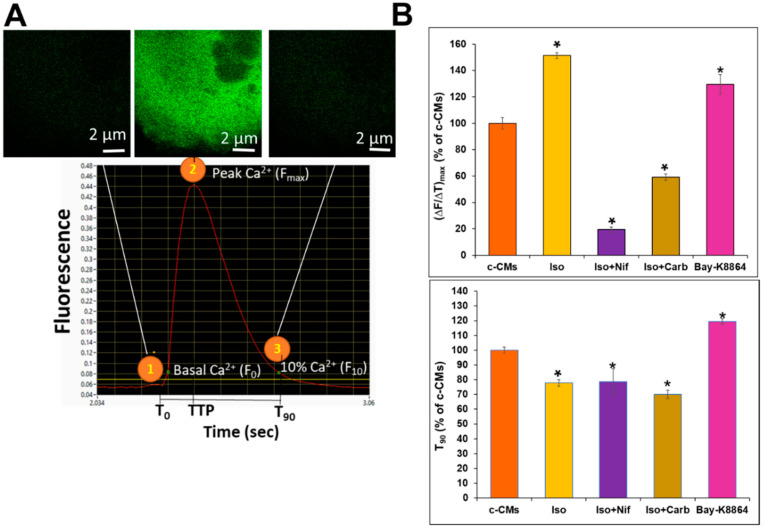
Effects of various agonists on [Ca^2+^]_i_ fluctuations during contraction of cardiomyocytes (CMs). (**A**), Induction of genetically encoded calcium indicator (GECI)-enhanced green fluorescent protein (eGFP^+^)-CMs was induced by adding Doxycycline (500 nM) for 6 h. Live-imaging of [Ca^2+^]_i_ fluctuations during contraction of GECI-eGFP^+^-control-CMs in the presence and absence of the distinct agents were captured with the Olympus FluoView1000 confocal system (50 fps; 10 to 30 s; 60× objective; Em/Exit: 488:510 nm; see also Appendix A). The video recordings of the [Ca^2+^]_i_ transient fluctuations were analyzed with the software Video Analyzer 1.9, allowing determination of all the experimental parameters between the time points 1 to 3 in the figure (for the control CMs; c-CMs). Parameters were used to calculate the Time-to-peak (TTP), (∆F/∆T)_max_ and (time to 90% of peak)T_90_ in the presence and absence of different drugs. (**B**), Diagrams show the effects of the different agonists and antagonists on (∆F/∆T)_max_ and T_90_ values of the Ca^2+^ transient. Values are expressed as a percent of the c-CM values, which were set to 100% (mean ± SEM, *n* = 6, * *p* < 0.05; 6 independent experiments).

**Figure 4 cells-11-01280-f004:**
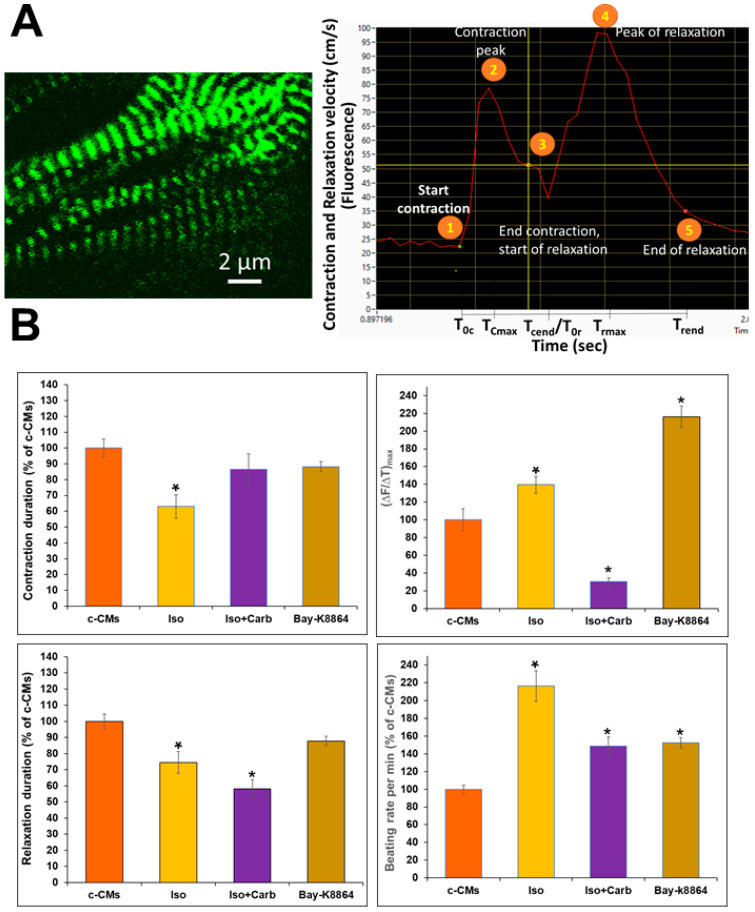
Live-imaging of contraction and relaxation velocity activity of α-actinin (ACTN2)-copepod green fluorescent protein (copGFP^+^)-cardiomyocytes (CMs). Live-imaging of the contractile and relaxation velocities of the ACTN2-copGFP^+^-CMs in the presence and absence of the distinct agents. Video recordings were captured with the Olympus FluoView1000 confocal system (50 fps; 10 to 30 s; 60× oil objective). (**A**), ACTN2 is enriched in Z-discs of the sarcomeres of ACTN2-copGFP^+^-CMs (see also Appendix A). The video recordings of the fluctuations of the contractile and relaxation velocities were analyzed with the software Video Analyzer 1.9, allowing determination of all the experimental parameters between the time points 1 to 5 in the figure (for control CMs; c-CMs). Parameters were used for the calculation of time-to-peak (TTP), slope (∆F/∆T)_max_, T_90_, and the contraction and relaxation times for c-CMs. (**B**), Diagrams show the effects of the different agonists and antagonists on (∆F/∆T)_max_, on the contraction/relaxation times and on the beating frequency in the presence and absence of the different drugs. Values are expressed as a percentage of the c-CM values, which were set to 100% (mean ± SEM, *n* = 6, * *p* < 0.05; 6 independent experiments).

## Data Availability

Not applicable.

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
