# Peer review of "Live-Cell Imaging of the Contractile Velocity and Transient Intracellular Ca2+ Fluctuations in Human Stem Cell-Derived Cardiomyocytes"

_cells, 2022, doi:10.3390/cells11081280_

Round 1

Reviewer 1 Report

This well-written manuscript describes the development of two live-cell imaging techniques to monitor, via biosensors, the effects of drugs on the function of human cardiomyocytes (CMs) in real-time. For live-cell imaging of transient alterations of [Ca2+]i, the authors engineered human induced pluripotent stem cells (hiPSCs) that carry a genetically-encoded Ca2+-indicator (GECI). They also engineered hiPSCs expressing an alpha-cardiac actinin (ACTN2)-copepod (cop) green fluorescent protein. This cell line was created using the CRISPR-Cas9 system with a homology directed recombination approach. These cell lines were then induced to become CMs expressing their respective biosensors. Using these induced CMs, the authors could measure not only Ca2+ transients, but also sarcomere shortening and relaxation in real-time by monitoring the relative motion of the Z-discs. Finally, the authors developed a video analysis software tool to quantify sarcoplasmic Ca2+ fluctuations in CMs and contractions of CMs. The authors used several standard drugs with well-known ionotropic, lusitropic and chronotropic effects on CMs to validate their cell lines and software. This novel platform will doubtless facilitate high-throughput drug screenings in human CMs. I have only a few minor comments:

Minor Comments:

  1. Legend to Figure 1: Please indicate meaning of attL, attB, attR and indicate in the legend the role of the pCAGPBase plasmid (transposon vector).
  2. Be consistent with spelling of puromycin.
  3. Define TTP and T90 in legend to Figure 2.

Author Response

Response: We would like to thank reviewer 1 for the very positive comments.Minor Comments:

  1. Legend to Figure 1: Please indicate meaning of attL, attB, attR and indicate in the legend the role of the pCAGPBase plasmid (transposon vector). Done

Response: Done as suggested (see Figure 1 legend)

  1. Be consistent with spelling of puromycin. Done

        Response: Done as suggested

  1. Define TTP and T90 in legend to Figure 2. Done

Response: Done as suggested (see in legend to Figure 2)

Reviewer 2 Report

The manuscript by Acharya et al. reports establishment of two human iPS cell lines carrying GCaMP Ca2+ sensing protein or GFP-fused cardiac actinin. They integrated the transgene into the genome or directly edited the genome by CRISPR technique. These cell lines are then differentiated into cardiomyocytes to study cardiac calcium signaling and contraction. Despite this elegant technic, however, the authors fail to show any novel physiological findings. I have also some concerns on the novelty of the study.

Human stem cell lines carrying GCaMP have been already reported (e.g. Zhu et al. (2014) Methods Mol Biol 1181, 229-247; Jiang et al. (2018) Science 9, 27-35). What is the novelty of the present study? Failure to cite these papers does not make the method novel! The same concern applies also for the cardiac contractile protein with reporter gene. Ribeiro et al (JMCC (2020) 141, 54-64) generated human ES cell line expressing alpha-actinin with a reporter gene, and measured cardiomyocyte contraction. Authors’ measurements including the video microscopy and the analyzing program may be novel when combined with the genetically engineered stem cell line, but I find no novel or interesting finding in their reported data. The manuscript should point out and discuss what are  the new findings and compare their data to those reported by others. The study could also benefit from attempts to mature the cardiomyocytes as has been reported by others.

Minor Points

Is cardiac actinin expression altered by CRISPR or fusing with GFP?

Please show representative traces (time courses) of Ca2+ transients and contraction in the presence and absence of drugs, as well as when rate staircase phenomenon is examined. Showing such tracings will allow the reader to better judge the quality and stability of the recordings and asses the maturity of cells used in this report.

Author Response

Reviewer 2

Comments and Suggestions for Authors II

The manuscript by Acharya et al. reports establishment of two human iPS cell lines carrying GCaMP Ca2+ sensing protein or GFP-fused cardiac actinin. They integrated the transgene into the genome or directly edited the genome by CRISPR technique. These cell lines are then differentiated into cardiomyocytes to study cardiac calcium signaling and contraction. Despite this elegant technic, however, the authors fail to show any novel physiological findings. I have also some concerns on the novelty of the study.

Human stem cell lines carrying GCaMP have been already reported (e.g. Zhu et al. (2014) Methods Mol Biol 1181, 229-247; Jiang et al. (2018) Science 9, 27-35). What is the novelty of the present study? Failure to cite these papers does not make the method novel! The same concern applies also for the cardiac contractile protein with reporter gene. Ribeiro et al (JMCC (2020) 141, 54-64) generated human ES cell line expressing alpha-actinin with a reporter gene, and measured cardiomyocyte contraction. Authors’ measurements including the video microscopy and the analyzing program may be novel when combined with the genetically engineered stem cell line, but I find no novel or interesting finding in their reported data. The manuscript should point out and discuss what are the new findings and compare their data to those reported by others. The study could also benefit from attempts to mature the cardiomyocytes as has been reported by others.

Response: We really appreciate the reviewer’s comments. Regarding the novelty of our research work presented in thus study, we have not claimed novelty for generating the GCaMP construct itself however our study differ significantly than the previous study (papers reference here by reviewer 2 and 3) in the way how the GCaMP donor vector is created and expressed in the hiPSCs. With the use of our construct, user can precisely control the expression of the GCaMP using Tet-on system. We also opted for directional insertion of the GCaMP in AAVS1 site however even after multiple attempts, we were not able to generate the lines. This is by no means the problem faced by our group only, as during the troubleshooting we found that several other groups also struggle to generate the ipsc lines using AAVS1 locus targeting. Our PigggBac based construct by-passes all those hurdles and we were able to generate the ipsc lines with relative ease. The ipsc lines generated maintain their pluoripotency and as shown in the study they can be efficiently differentiated into CMs.

Please consider that we cited the suggested references (see lines 365-371 and references ). Also, since the referee 3 also had almost the same critisisms we responded exacly in the same way. 

Response to the ES cell line expressing alpha-actinin with a reporter gene:  We agree with the reviewer here that using innovative techniques like CRISPR-Cas9 generation of the reporter line has become relatively easier. To date, several such lines have been reported, but the key advantage with our line is that we have edited the native ACTN gene by removing its stop codon and replacing it with the eGFP. This is very different from many other reported lines in terms that they have used the promoters from either ACTN or MYH genes to express eGFP. This method incorporates excess DNA material in the cellular genome that may falsify the results (see lines 397-403). Please notice that the referee 3 also raised this point; Therefore, the response remains the same.

Minor Points

Is cardiac actinin expression altered by CRISPR or fusing with GFP?

Response: No, CRISPR editing takes place at the STOP codon of ACTN2 gene which does not affect the promoter region hence does not affect the ACTN2 expression.

Please show representative traces (time courses) of Ca2+ transients and contraction in the presence and absence of drugs, as well as when rate staircase phenomenon is examined. Showing such tracings will allow the reader to better judge the quality and stability of the recordings and asses the maturity of cells used in this report. A supplementary fig can be made to incorporate representative traces.

Response: Supplementary figures were included (Figure S2 and S3, supplemental materials document)

Reviewer 3 Report

This article presents two iPS cell lines encoding fluorescent proteins for assessing either calcium signaling (GCaMP6) or contractility (ACTN2-GFP). It also presents a software package for analyzing the calcium and contractility dynamics from these lines. The authors then use this platform to show the effect of several drugs with known mechanisms of action, and show that the lines behave in the predicted fashion.

MAJOR CONCERNS:

  1. While these lines are useful, they largely duplicate tools that are already published. Notably, the claim that "we are the first group worldwide to generate genetically encoded Ca2+ biosensor/GCaMP hiPSC-CMs" overlooks the line described 7 years ago by Huebsch et. al. (PMID 26971820) that carries a GCaMP6f in the AAVS1 locus and was used to assess calcium transients in hiPSC-CMs. Similarly, the discussion that their ACTN2-GFP line has advantages over previous lines because it is a knockin rather than a transgenic insertion overlooks previous, near-identical ACTN2-GFP knockins (see Roberts et al, 2019 PMID 30956114), as well as the TTN-GFP knockin described in the Toepfer et al reference included in this manuscript.
  2. While the software package seems to function properly, and, per the authors, represents a user-friendly interface, the authors do not present any direct comparisons to other published methods. They do present some commentary on the limitations of other methods, but no direct comparison to show their method is at least performs equivalently to previously published methods. Additionally, some of the commentary on previously published analysis platforms is not accurate. For instance, the authors comment on the capabilities of the MUSCLEMOTION program, stating that "there is no mention of its use for single-cell analysis". However, this is expressly addressed as an application of that method in the paper describing this program (Sala et al 2018).

MINOR CONCERNS:

  1. Using GCaMP does indeed have advantages over dyes, but some of the stated advantages, such as the decrease in cell to cell variability due to dye uptake, may not be as significant, as one can also have variability in expression at baseline or in response to doxycycline.
  2. MUSCLEMOTION is an easy-to-use ImageJ plugin, while the text as written seems to suggest that it is a more difficult-to-use Matlab program.
  3. In Figure 3, "ACTN2" is written as "ACNT2"
  4. The plasmid containing GCaMP6s appears to be a viral plasmid given the presence of ITRs, but it also appears from the figure that this was simply transfected directly into iPS cells rather than going through viral packaging. Is this correct? If so then why use a viral plasmid, which contains many potentially detrimental viral sequences, rather than a more standard plasmid backbone?

Author Response

Reviewer 3

This article presents two iPS cell lines encoding fluorescent proteins for assessing either calcium signaling (GCaMP6) or contractility (ACTN2-GFP). It also presents a software package for analyzing the calcium and contractility dynamics from these lines. The authors then use this platform to show the effect of several drugs with known mechanisms of action, and show that the lines behave in the predicted fashion.

MAJOR CONCERNS:

  1. While these lines are useful, they largely duplicate tools that are already published. Notably, the claim that "we are the first group worldwide to generate genetically encoded Ca2+ biosensor/GCaMP hiPSC-CMs" overlooks the line described 7 years ago by Huebsch et. al. (PMID 26971820) that carries a GCaMP6f in the AAVS1 locus and was used to assess calcium transients in hiPSC-CMs. Similarly, the discussion that their ACTN2-GFP line has advantages over previous lines because it is a knockin rather than a transgenic insertion overlooks previous, near-identical ACTN2-GFP knockins (see Roberts et al, 2019 PMID 30956114), as well as the TTN-GFP knockin described in the Toepfer et al reference included in this manuscript.

Response: We really appreciate the reviewer’s comments. Regarding the novelty of our research work presented in thus study, we have not claimed novelty for generating the GCaMP construct itself however our study differ significantly than the previous study (papers reference here by reviewer 2 and 3) in the way how the GCaMP donor vector is created and expressed in the hiPSCs. With the use of our construct, user can precisely control the expression of the GCaMP using Tet-on system. We also opted for directional insertion of the GCaMP in AAVS1 site however even after multiple attempts, we were not able to generate the lines. This is by no means the problem faced by our group only, as during the troubleshooting we found that several other groups also struggle to generate the ipsc lines using AAVS1 locus targeting. Our PigggBac based construct by-passes all those hurdles and we were able to generate the ipsc lines with relative ease. The ipsc lines generated maintain their pluoripotency and as shown in the study they can be efficiently differentiated into CMs. Please consider that we cited the suggested references (see lines 365-371 and references ). Also, since the referee 2 also had almost the same critisisms we responded exacly in the same way. 

Response: (ACTN2-GFP line)

We agree with the reviewer here that using innovative techniques like CRISPR-Cas9 generation of the reporter line has become relatively easier. To date, several such lines have been reported, but the key advantage with our line is that we have edited the native ACTN gene by removing its stop codon and replacing it with the eGFP. This is very different from many other reported lines in terms that they have used the promoters from either ACTN or MYH genes to express eGFP. This method incorporates excess DNA material in the cellular genome that may falsify the results (see lines 397-403).

  1. While the software package seems to function properly, and, per the authors, represents a user-friendly interface, the authors do not present any direct comparisons to other published methods. They do present some commentary on the limitations of other methods, but no direct comparison to show their method is at least performs equivalently to previously published methods. Additionally, some of the commentary on previously published analysis platforms is not accurate. For instance, the authors comment on the capabilities of the MUSCLEMOTION program, stating that "there is no mention of its use for single-cell analysis". However, this is expressly addressed as an application of that method in the paper describing this program (Sala et al 2018).

Response:

Thank you very much for the constructive comment.

Regarding the software tool presented in this study, the major advantages of our tools are,

  1. The user does not have to download any additional plug-ins or software components separately. It is an all in one tool and has all the necessary software components built into it. 2. User can use any windows based operating system to use the software and no specific hardware ware upgrade is needed. 3. With this tool user can analyze CMs calcium flux and CMs beating simultaneously hence saving lot of time. 4. This tool is most suitable for high through put single cell analysis as well (see discussion, line 406-419)

MINOR CONCERNS:

  1. Using GCaMP does indeed have advantages over dyes, but some of the stated advantages, such as the decrease in cell to cell variability due to dye uptake, may not be as significant, as one can also have variability in expression at baseline or in response to doxycycline.

Response: This is true, however to minimize this expression variability we did clonal selection after the puromycin selection and selected the clone which showed uniform expression pattern (data not shown in this manuscript).

  1. MUSCLEMOTION is an easy-to-use ImageJ plugin, while the text as written seems to suggest that it is a more difficult-to-use Matlab program.

Response: Paragraph modified as shown in the text

  1. In Figure 3, "ACTN2" is written as "ACNT2"

Response: Done as suggested

  1. The plasmid containing GCaMP6s appears to be a viral plasmid given the presence of ITRs, but it also appears from the figure that this was simply transfected directly into iPS cells rather than going through viral packaging. Is this correct? If so then why use a viral plasmid, which contains many potentially detrimental viral sequences, rather than a more standard plasmid backbone?

Response: The ITR (Inverted Terminal Repeat) regions corresponds to the PiggyBac transposase system which was used to generate the stable lines. Yes, the plasmids were transfected using magnetofection and without use of viral particles.

Reviewer 4 Report

The manuscript entitled with “Live-time imaging of the contractile velocity and transient intracellular Ca2+ fluctuations in human stem cell-derived cardiomyocytes” proposes the employment of a house-made software to analyse the live cell dynamics in real-time. This is a very good work showing that live-time imaging can be very useful and give interesting results in cardiomyocyte contraction/relaxation kinetics under different relevant conditions. The launched software presents evident improvements compared to other already available programs (like SarcOptim or CalTrack). The refined results obtained are very well discussed. The manuscript is well organized and well written. In my opinion the present manuscript is approximately acceptable to the publication in Cells except for the following minor points:

-------------

The Introduction section contains some repetitions which should be avoided like “We firmly believe that these sophisticated tools open new horizons to study the contractility of CMs under pathological conditions. These tools can be applied to ….” (lines 82-84). Please, I may ask to the authors to change the second term “These tools” by “This technology”.

-------------

In the Materials and Methods section many abbreviations concerning chemical compounds should be detailed. For example, I may modify “containing 10 µM IWP2 (Tocris, United Kingdom) (line 102) by the following statement:

“containing 10 µM of N-(6-Methyl-2-benzothiazolyl)-2-[(3,4,6,7-tetrahydro-4-oxo-3-phenylthieno[3,2-d]pyrimidin-2-yl)thio]-acetamide (IWP2, Tocris, United Kingdom)”.

Authors introduce the medium “Roswell Park Memorial Institute (RPMI; Thermo Fisher, 106 MA, US) (lines 106-107)” but it should be referred before in the line 98 (“Then, cells were cultured in RPMI1640/B-27 without …”). 

Other example is the “generation of the genetically encoded Ca2+ indicator (GECI)”. Even if it is defined in the abstract section, it would be referred again on the M&M section: “we engineered hiP-124 SCs carrying a GECI” (lines 124-125) instead of the results section: “The generation of the genetically encoded Ca2+ indicator (GECI)-enhanced green fluorescent protein (eGFP+)-hiPSCs (lines 196-197). In this case, the GECI definition appeared on the legend of Figure 1 may be erased.

-------------

Figure 2B (downward part, regarding T90 values of the Ca2+ transient) lacks the standard deviation of c-CMs bar. Please, check this aspect and include the SD as the upper Figure 2B.

The authors also should take care of the provided significant digits being consistent for the entire Results section. For example, “…. increase by the CM beating rate by 2.2- and 1.4-fold, respectively, compared to that of c-CMs. In contrast, consecutive treatments with Isoprenaline and Carbachol resulted in only a 1.48-fold increase in the CM beating rate …” (lines 320-322). I may suggest to change the second term by adding 1.5-fold instead 1.48-fold.   

Author Response

Referee 4:

The manuscript entitled with “Live-time imaging of the contractile velocity and transient intracellular Ca2+ fluctuations in human stem cell-derived cardiomyocytes” proposes the employment of a house-made software to analyse the live cell dynamics in real-time. This is a very good work showing that live-time imaging can be very useful and give interesting results in cardiomyocyte contraction/relaxation kinetics under different relevant conditions. The launched software presents evident improvements compared to other already available programs (like SarcOptim or CalTrack). The refined results obtained are very well discussed. The manuscript is well organized and well written. In my opinion the present manuscript is approximately acceptable to the publication in Cells except for the following minor points:

The Introduction section contains some repetitions which should be avoided like “We firmly believe that these sophisticated tools open new horizons to study the contractility of CMs under pathological conditions. These tools can be applied to ….” (lines 82-84). Please, I may ask to the authors to change the second term “These tools” by “This technology”.

Response: Done as suggested line 84

In the Materials and Methods section many abbreviations concerning chemical compounds should be detailed. For example, I may modify “containing 10 µM IWP2 (Tocris, United Kingdom) (line 102) by the following statement:

“containing 10 µM of N-(6-Methyl-2-benzothiazolyl)-2-[(3,4,6,7-tetrahydro-4-oxo-3-phenylthieno[3,2-d]pyrimidin-2-yl)thio]-acetamide (IWP2, Tocris, United Kingdom)”.

Authors introduce the medium “Roswell Park Memorial Institute (RPMI; Thermo Fisher, 106 MA, US) (lines 106-107)” but it should be referred before in the line 98 (“Then, cells were cultured in RPMI1640/B-27 without …”). 

Response: Done as suggested (line 99,104,119,120,123,140,266,268,274,279)

Other example is the “generation of the genetically encoded Ca2+ indicator (GECI)”. Even if it is defined in the abstract section, it would be referred again on the M&M section: “we engineered hiP-124 SCs carrying a GECI” (lines 124-125) instead of the results section: “The generation of the genetically encoded Ca2+ indicator (GECI)-enhanced green fluorescent protein (eGFP+)-hiPSCs (lines 196-197). In this case, the GECI definition appeared on the legend of Figure 1 may be erased.

Response: -done as per suggested (line number 132,219)

-------------

Figure 2B (downward part, regarding T90 values of the Ca2+ transient) lacks the standard deviation of c-CMs bar. Please, check this aspect and include the SD as the upper Figure 2B.

Response: Done as suggested

The authors also should take care of the provided significant digits being consistent for the entire Results section. For example, “…. increase by the CM beating rate by 2.2- and 1.4-fold, respectively, compared to that of c-CMs. In contrast, consecutive treatments with Isoprenaline and Carbachol resulted in only a 1.48-fold increase in the CM beating rate …” (lines 320-322). I may suggest to change the second term by adding 1.5-fold instead 1.48-fold. 

Response: -done as per suggested line number 351

Round 2

Reviewer 3 Report

The authors have nicely clarified the novelty and use of their line in respect to the previous literature, specifically highlighting the inducibility of the GCAMP expression, which I agree is a potential advantage. While the ACTN2-GFP line is very similar to other constructs that have knocked in GFP to sarcomeric genes, it is valuable that it is in a new genetic background (which could be highlighted), and combined with their software package it does represent a nice toolset. Overall the authors present a nice combination of tools and drug validation.

Minor comments:

1) I notice now that loxP sites are incorporated in the ACTN2-GFP line. Were these subsequently removed using Cre recombinase expression, or is this promoter/construct still incorporated in the locus? If it is still there, it has the potential to disrupt faithful expression (although the author's data does look good here).

2) Are the c-CM controls separate cells or matched pre-treatment? If they are controls, were they treated with the carrier for the compounds (e.g. DMSO) or were they untreated controls?

Author Response

1) I notice now that loxP sites are incorporated in the ACTN2-GFP line. Were these subsequently removed using Cre recombinase expression, or is this promoter/construct still incorporated in the locus? If it is still there, it has the potential to disrupt faithful expression (although the author's data does look good here).

Response: No, we did not remove the LOX site by the CRE recombinase. In our previous experiments, we removed the LoX site. However, we faced some differentiation problems with the LOX- hiPSCs since the differentiation of these cells to cardiomyocytes was less effective.  Therefore we kept  Lox site in hiPSCs which resulted in an effective differentiation into ACTN2 CMs (the purity was higher than > 90%).

2) Are the c-CM controls separate cells or matched pre-treatment? If they are controls, were they treated with the carrier for the compounds (e.g. DMSO) or were they untreated controls?

Response: Response: Yes, the control c-CMs were treated with 0.05% DMSO (see page 7, lines 4 and 5)